# High Tumor-Infiltrating Lymphocyte Count Is Associated with Distinct Gene Expression Profile and Longer Patient Survival in Advanced Ovarian Cancer

**DOI:** 10.3390/ijms241813684

**Published:** 2023-09-05

**Authors:** Andras Jozsef Barna, Zoltan Herold, Miklos Acs, Sandor Bazsa, Jozsef Gajdacsi, Tamas Marton Garay, Magdolna Herold, Lilla Madaras, Dorottya Muhl, Akos Nagy, Attila Marcell Szasz, Magdolna Dank

**Affiliations:** 1Department of Obstetrics and Gynecology, Saint Pantaleon Hospital, H-2400 Dunaujvaros, Hungary; 2Division of Oncology, Department of Internal Medicine and Oncology, Semmelweis University, H-1083 Budapest, Hungary; 3Department of Surgery, University Hospital, D-93053 Regensburg, Germany; 4Directorate General of Medical Quality Assurance, Semmelweis University, H-1085 Budapest, Hungary; 5Faculty of Information Technology and Bionics, Pazmany Peter Catholic University, H-1083 Budapest, Hungary; 6Department of Internal Medicine and Hematology, Semmelweis University, H-1088 Budapest, Hungary; 7Department of Pathology, Forensic and Insurance Medicine, Semmelweis University, H-1091 Budapest, Hungary; 8Department of Pathology and Experimental Cancer Research, Semmelweis University, H-1085 Budapest, Hungary

**Keywords:** ovarian neoplasms, immunohistochemistry, NanoString, CD4-positive T-lymphocytes, CD8-positive T-lymphocytes, CD45-positive leukocytes, leukocyte common antigens

## Abstract

Cancer-related immunity plays a significant role in the outcome of ovarian cancer, but the exact mechanisms are not fully explored. A retrospective, real-life observational study was conducted including 57 advanced ovarian cancer patients. Immunohistochemistry for CD4^+^, CD8^+^, and CD45^+^ was used for assessing tumor-infiltrating immune cells. Furthermore, an immune-related gene expression assay was performed on 12–10 samples from patients with less than and more than 1-year overall survival (OS), respectively. A higher number of CD4^+^ (*p* = 0.0028) and CD45^+^ (*p* = 0.0221) immune cells within the tumor microenvironment were associated with longer OS of patients. In a multivariate setting, higher CD4^+^ T cell infiltration predicted longer OS (*p* = 0.0392). Twenty-three differentially expressed genes—involved in antigen presentation, costimulatory signaling, matrix remodeling, metastasis formation, and myeloid cell activity—were found when comparing the prognostic groups. It was found that tumor-infiltrating immune cell counts are associated with peculiar gene expression patterns and bear prognostic information in ovarian cancer. *SOX11* expression emerged and was validated as a predictive marker for OS.

## 1. Introduction

Ovarian cancer has an incidence and mortality rate of over 300,000 new cases and 200,000 deaths per year, respectively. It is the 8th most common cancer in females, based on the 2020 GLOBOCAN data [1]. The early symptoms of the disease are not alarming; thus, the majority (over 70%) of ovarian cancer patients are diagnosed at an advanced stage, where the disease has spread throughout the pelvis or elsewhere in the abdomen with or without the involvement of lymph nodes [2,3]. Tumors originating from an epithelial origin are the most common (~95%) and include the following four primary histological subtypes: serous, endometrioid, mucinous, and clear cell ovarian cancer [3].

In the last decades, cancer-related immunity has become a particularly important research area due to its complex relationship with tumor initiation, growth, and progression [4]. Consequently, considerable research interest is focused on the tumor microenvironment (TME). This includes the identification of possible molecular targets, which are usually involved in cancer-related inflammation [5,6]. Similar to other cancer types [7], ovarian cancer can also be characterized by tumor-infiltrating immune cells and cancer-specific cytokines, which have been proven to be promising prognostic markers. However, their exact role in progression and their relationship with the different immune components have not been fully understood [6,8,9,10]. The results of recent studies suggest that the higher number of CD4^+^, CD8^+^, and/or CD45^+^ immune cells in the TME are associated with improved survival in patients with solid tumors [6,11].

Although several studies investigated gene profiles of ovarian cancer patients in various comparisons [11,12,13,14,15,16,17], to our knowledge, no previous study investigated whether the number of tumor-infiltrating immune cells is associated with any gene expression differences. Therefore, a retrospective, real-life observational study (RROS) was conducted, where a detailed analysis of selected ovarian cancer samples was performed to identify a relevant association between the different tumor-infiltrating immune cells and clinicopathological parameters, and their impact on prognosis. Furthermore, the secondary goal of the RROS was to investigate whether there are differences in the various immune functions of those patients with different tumor-infiltrating immune cells, and what gene expression differences could be detected.

## 2. Results

A total of 57 patients with primary ovarian cancer were included in this RROS. Formalin-fixed and paraffin-embedded (FFPE) samples were obtained from all participants. CD4^+^, CD8^+^, and CD45^+^ tumor-infiltrating immune cells were analyzed in all tumor samples. Furthermore, a subset of 22 samples (12 with poor and 10 with excellent prognosis) was also investigated using NanoString assay-based gene expression profiling. Anamnestic, clinical, and histopathological data of all study subjects are summarized in Table 1. Data of the oncological treatments could be collected for 44 patients (77.2%). The remaining patients were treated at other institution(s) due to their preference. Two and five patients received no and the best supportive treatment only, due to stage I and advanced disease, respectively. Paclitaxel with carboplatin/cisplatin, pegylated liposomal doxorubicin, and gemcitabine + carboplatin with or without bevacizumab were used as the most common first- and second/third-line treatments. Moreover, topotecan, etoposide, and olaparib were used as late-line options.

### 2.1. CD4^+^, CD8^+^, and CD45^+^ Immunohistochemical Analysis of Ovarian Tumor Samples

FFPE samples were immunostained for CD4^+^, CD8^+^, and CD45^+^ tumor-infiltrating immune cells, where 9, 15, 17, and 16; 7, 24, 14, and 12; and 3, 10, 10, and 34 samples were stained negatively, weakly, moderately, and strongly for CD4^+^, CD8^+^, and CD45^+^ tumor-infiltrating immune cells, respectively (Figure 1). Of the 57 study participants, 3 (5.3%) were negative for all three immune cells, and 10 and 44 patients had immune cell positivity for 2 and 3 markers, respectively. It was investigated whether any of the clinical and/or histopathological parameters affected the staining categories. It was found that those patients with moderate (59.70 ± 13.21 years) and strong (57.51 ± 16.17 years) CD4^+^ staining were significantly younger (Kruskal–Wallis *p* = 0.0355), compared to those with weak staining (70.97 ± 16.00 years; Figure 2A). Those patients with negative and weak CD8^+^ staining had type 2 diabetes mellitus in their medical history more often than those within the other two categories (negative: 57.14%; weak: 37.50%; moderate: 7.14%; strong: 8.33%; *p* = 0.0194). Moreover, both hemoglobin (Figure 2B) and hematocrit (Figure 2C) values were tendentiously lower in the strongly stained CD8^+^ cohort. Furthermore, the length of hospitalization was shorter with the increase of CD45^+^ leukocyte staining (Kruskal–Wallis *p* = 0.0346; Figure 2D). No further differences and/or relationships could be justified.

It was also ©nvestigated whether there is any correlation between the various tumor-infiltrating immune cells and other clinical parameters. No connection could be found between CD4^+^ and CD8^+^ (*p* = 0.9157). A significant connection was observed between CD45^+^ and CD4^+^ (Spearman’s rho (ρ): +0.72; *p* < 0.0001), and between CD45^+^ and CD8^+^ (ρ: +0.51; *p* < 0.0001). Furthermore, negative correlations were found between CD4^+^ and age (ρ: −0.27; *p* = 0.0480), CD8^+^ and hemoglobin (ρ: −0.28; *p* = 0.0394), and CD8^+^ and hematocrit (ρ: −0.27; *p* = 0.0407). No further significant correlations could be justified.

Survival analyses of raw CD4^+^, CD8^+^, and CD45^+^ tumor-infiltrating immune cell percentages revealed that the higher the CD4^+^ (*p* = 0.0028) and the CD45^+^ (*p* = 0.0221) infiltration within the FFPE samples was, the longer the overall survival (OS) of the patients. However, no such connection could be justified in the case of CD8^+^ staining (Table 2). When investigating the results in a multivariate setting, where the three immune markers were analyzed together, only the CD4^+^ results significantly predicted the OS of patients (*p* = 0.0392; Table 2). In addition to the raw numerical values, we also examined OS in the CD4^+^, CD8^+^, and CD45^+^ sub-cohorts. As above, we could observe that patients with higher staining categories usually had improved OS (Figure 3 and Table 3).

Furthermore, a second multivariate survival model with additional clinical parameters was also created, where in addition to the CD4^+^, CD8^+^, and CD45^+^ immune cell percentages, age, hemoglobin levels, platelet counts, the duration of hospitalization, histology, and the American Society of Anesthesiologists (ASA) performance scores were also added to the model as explanatory variables. The beneficial effect of higher CD4^+^ percentages (*p* = 0.0286) could be observed in this model as well, in addition to the worsening survival effects of higher platelet counts (*p* = 0.0077), longer hospitalization durations (*p* = 0.0054), and higher ASA scores (Appendix A).

### 2.2. Immune Panel Gene Expression Analysis

A total of 22 (38.60%) FFPE samples were selected for analysis with the NanoString nCounter^®^ PanCancer IO 360^TM^ Gene Expression Panel. The samples were selected based on the survival times of patients, where 12–10 had an OS of less than or more than 1 year (median survival times: 4.37 months vs. not reached). The clinical characteristics of the two groups were compared. A statistical difference was found in the CD4^+^ (*p* = 0.0038) and CD45^+^ (*p* = 0.0465) tumor-infiltrating immune cell percentages, platelet count (*p* = 0.0408), and ASA performance scores (*p* = 0.0217) of the patients. In all cases, the clinically worse values were associated with the poor prognosis group. The ASA performance scores of the two groups were inversely proportional to each other. Moreover, the length of hospitalization was marginally shorter in the good prognosis group (*p* = 0.0565; Appendix A). Furthermore, the number of patients who received the best supportive care therapy or a single chemotherapy regimen only with a lower number of cycles, was higher in the poor prognosis group. In contrast, in the good prognosis group, several lineages of chemotherapy were characteristic.

Differentially expressed genes (DEGs) were identified using differential expression analysis. Of the 770 tested genes, 23 significant DEGs were found when comparing the prognosis groups (Figure 4). Of those 23 DEGs, 10 and 13 were up- and down-expressed, respectively. The complete list of DEGs, including their functional and cancer-immunity cycle annotations, can be read in Appendix A. Gene over-representation was investigated via gene set enrichment analysis (GSEA). Here, 5, 6, 5, and 10 of the 23 DEGS were involved in antigen presentation, costimulatory signaling, matrix remodeling and metastasis formation, and in myeloid cell activity, respectively. GSEA results suggested that, except for matrix remodeling and metastasis, all pathways were downregulated in those ovarian cancer patients with worse survival (Appendix A).

Expression patterns were further investigated based on the CD4^+^, CD8^+^, and CD45^+^ tumor-infiltrating immune cell categories, ASA, and on tumor extent (T status). It must be noted that due to the number of smaller sample sizes that could be included in the NanoString analyses, some previously independent categories had to be combined: 12, 21, and 12 DEGs were found in the ‘CD4^+^ 0%’ vs. ‘CD4^+^ 1–5%’, ‘CD4^+^ 0%’ vs. ‘CD4^+^ 5%<’, and ‘CD4^+^ 1–5%’ vs. ‘CD4^+^ 5%<’ comparisons, respectively (Appendix A). While GSEA revealed no differences in the latter comparison, in the former two, myeloid compartment, matrix remodeling and metastasis, and common signaling pathway functions were down-expressed, and immunometabolism, the killing of cancer cells, and myeloid cell activity functions were up-expressed in those patients with higher CD4^+^ T cell infiltrations (Appendix A). Six and thirteen significant DEGs were found between the negatively/weakly vs. moderately CD8^+^-stained (Appendix A) and the moderately vs. strongly CD8^+^-stained patient groups (Appendix A), respectively. However, no significant DEGS could be identified in the case of the negatively/weakly vs. strongly CD8^+^-stained cohorts (Appendix A). While most DEGs of the first CD8^+^ comparison affected immune cell adhesion, migration, and localization to tumors, in the second comparison, most DEGs were associated with lymphoid compartment, myeloid cell activity, and immune cell localization to tumors’ functions (Appendix A). Finally, 19 significant DEGs were found when comparing the negatively/weakly/moderately vs. strongly CD45^+^-stained cohorts (Appendix A). Of these, the killing of cancer cells, myeloid cell activity, T cell priming and activation, and cell cycle and proliferation functions were up-expressed, and the myeloid compartment function was down-expressed in the strongly stained cohort (Appendix A).

When comparing ASA groups I vs. II, I vs. III, and II vs. III, 13, 26, and 37 DEGs were found, respectively. No gene was found that was significant in all three comparisons (Appendix A). The following pathways were affected by the ASA score groups. The genes belonging to antigen presentation (ASA I vs. II) and immune cell adhesion and migration (ASA I vs. II) were all up-expressed, and myeloid cell activity (ASA II vs. III) was down-expressed. Half of the genes associated with myeloid compartment (ASA II vs. III) were both down- and up-expressed. Stromal factors were up-expressed in the ASA II group, compared to the other two ASA performance score groups (Appendix A). The effect of tumor extent (T status) was also investigated. Here, 41 DEGs were found when T status I and II–III were compared (Appendix A), and the down- and up-expression of matrix remodeling and metastasis and common signaling pathways could be observed, respectively (Appendix A). A heatmap summarizing all the above-detailed comparisons can be seen in Figure 5 and Appendix A.

In all the above-detailed comparisons, a total of 148 significant DEGs were identified, and 101 (68.24%), 27 (18.24%), 12 (8.11%), 6 (4.05%), and 1 (0.68%) of the 148 DEGs were associated with 1, 2, 3, 4, and 5 comparisons, respectively (Appendix A). The single gene associated with 5 comparisons was the tumor necrosis factor receptor superfamily member 10c (*TNFRSF10C*), and those associated with at least 4 comparisons are listed in Table 4. Based on their annotation details, 6 of these 7 DEGs were associated with myeloid cell activity (*BATF3*, *COL11A1*, *IL6*, *MMP1*, *PDZK1IP1*, and *TNFRSF10C*), 4 with common signaling pathways (*BATF3*, *COL11A1*, *IL6*, and *SOX11*), and 3 each with the killing of cancer cells (*IL6*, *SOX11*, and *TNFRSF10C*) and myeloid compartment pathways (*COL11A1*, *MMP1*, and *PDZK1IP1*). Those 12 DEGs that were associated with 3 comparisons (*ADM*, *CCL8*, *COL17A1*, *COMP*, *CST2*, *HLA-DQB1*, *HLA-DRB5*, *IL12RB2*, *PLA2G2A*, *PNOC*, *SBNO2*, *TPSAB1/B2*, and *ZC3H12A*) were associated most often with myeloid cell activity (7 of the 12: *CCL8*, *COL17A1*, *HLA-DQB1*, *HLA-DRB5*, *IL12RB2*, *SBNO2*, and *ZC3H12A*) and with immune cell localization to tumors (6 of the 12: *CCL8*, *HLA-DQB1*, *HLA-DRB5*, *IL12RB2*, *PNOC*, and *TPSAB1/B2*).

### 2.3. Comparison of NanoString Data with the “Kaplan–Meier Plotter” Web Application

To validate our data, it was investigated whether those DEGs with an FDR-adjusted *p* < 0.05, and ≥ or ≤±3 log_2_ fold-change, had the same effects over patient survival as detailed in the Kaplan–Meier Plotter web application (KMplotter [19]: https://kmplot.com/analysis/index.php?p=service&cancer=ovar; access date: 20 July 2023). We examined whether the hazard rates and median survivals obtained from our models were comparable to those observations verified in the KMplotter application [19]. It is notable that the KMplotter database contains only data of selected ovarian cancer patients with serous- and endometrioid-type tumors. Therefore, 13 of the 22 patients remained in this analysis, whose NanoString data were available and who had serous- or endometrioid-type ovarian cancer. To achieve sufficient statistical power in our survival models, the patients were divided into two groups based on the median of the RUVSeq normalized count data for each tested gene. A total of 17 DEGs were found, which met the above conditions (Table 5): *COL11A1*, *COL17A1*, *COMP*, *CTAG1B*, *HLA-DQA1*, *HLA-DQB1*, *HLA-DRB5*, *IL6*, *ITGB3*, *LYZ*, *MAGEC2*, *MMP1*, *PDZK1IP1*, *PLA2G2A*, *SOX2*, *SOX11*, and *TNFRSF18*. It was found that the over-expression of *SOX11* had a significant effect over patient survival in both datasets (this study: *p* = 0.0032; KMplotter: *p* = 0.0008). In addition, similar tendencies were found in the hazard rates and median survival times in the case of *COL11A1*, *COMP* and *PDZK1IP1*; however, their significant effect over survival could only be justified in the literature data [19] (Table 5).

## 3. Discussion

Tumor-infiltrating immune cells are characteristic of most solid tumors [20,21], including ovarian cancer [6,8]. Without being exhaustive, the following are known in the literature. CD4^+^ and CD8^+^ tumor-infiltrating lymphocytes (TILs) have been associated with improved OS and progression-free survival (PFS) in all epithelial ovarian cancers [11,22,23,24,25,26,27], including the high-grade serous [28,29], mucinous [25], clear cell [30], and endometrioid [31] types. CD8^+^ TILs had been found as a predictive marker for both 5-year, 10-year, and longer survival times [32,33]. Several studies have also investigated the spatial differences in these tumors. In mucinous ovarian carcinomas, CD4^+^ and CD8^+^ TILs were expressed at higher levels in the stroma compared to the epithelium [17]. Intraepithelial CD4^+^ TILs have been associated with an increased PFS and OS in the high-grade serous type [34]. Intra-islet CD4^+^ TILs of omental metastases were the main risk factors associated with poorer survival in advanced epithelial ovarian cancer, and the highest infiltration was found in the stroma of the omental tissue. CD4^+^ was significantly elevated in patients with FIGO stages IIIC/IV, compared to those with FIGO stages IIIA/IIIB [27]. Intraepithelial CD8^+^ TILs and a high CD8^+^/T-regulatory cell (T_reg_) ratio were associated with favorable prognosis [34,35]. Based on the results of a meta-analysis, intraepithelial CD8^+^ TILs have been associated with improved PFS, DFS, and OS [36].

It must be mentioned, however, that the opposite of the above was also reported in several studies. For example, the presence of stromal CD4 ^+^ and CD8 ^+^ TILs were found to be more characteristic for advanced-stage patients and disease-free survival (DFS), and OS was significantly shorter in the presence of CD8^+^ TILs [37]. The survival benefit of CD4^+^ TILs could not be justified in the study of Sato et al. [35]. High CD4^+^CD25^+^ T_reg_ infiltration has been associated with higher tumor grades, advanced stage, and suboptimal debulking, but not with survival [26]. In advanced clear cell ovarian cancer, an increased infiltration of CD4^+^ T cells at the leading edge and stroma was significantly associated with poorer OS [30]. CD8^+^ TILs were significantly higher in those patients with high-grade tumors, advanced-stage tumors, and omental metastasis, all with shorter DFS and poor prognosis [38]. The ratios of CD8^+^ TILs to CD4^+^CD25^+^ FOXP3^+^ and FOXP3^-^ T cells correlate with poor clinical outcomes; moreover, the authors of the study highlighted that the association found that the effector/suppressor ratios may be more important indicators of the disease outcome than individual cell counts [39].

Oncotherapy response has been the subject of significant research and clinical interest. In breast cancer, several studies have shown that a higher presence of TILs, particularly in triple-negative breast cancer (TNBC) and HER2-positive subtypes, is associated with a better response to chemotherapy and targeted therapies. For instance, the “IMpassion130” trial demonstrated improved OS and PFS in patients with advanced TNBC, who received atezolizumab and nab-paclitaxel, especially in those with high levels of TILs [40]. TILs have shown promise as predictive markers in ovarian cancer. Studies have indicated that higher levels of TILs, especially CD8^+^ T cells, are associated with improved responses to chemotherapy and immunotherapies [41]. These are just a few examples of how TILs have been investigated as predictive biomarkers in various cancer types. The presence and composition of TILs in the tumor microenvironment have been associated with improved responses to a range of oncotherapies, including chemotherapy, targeted therapies, and immunotherapies. However, it is essential to note that the predictive value of TILs may vary depending on the cancer type, stage, and treatment regimen, and ongoing research continues to refine our understanding of their role in predicting therapeutic responses. PARP-inhibitors, for instance, have been utilized mainly in breast and ovarian cancer, and there are no such convincing analyses published about TILs being predictive to date; however, their functional role has been extensively studied [42].

In addition, optimal cytoreduction and p53 mutations have been reported more often in those patients with more evident tumor-infiltrating lymphocyte staining [43]. A large-scale study with 3196 high-grade serous ovarian carcinoma patients revealed that CD8^+^ TILs were associated with longer patient survival regardless of the extent of the residual disease following cytoreduction, the type of oncological treatment, and the germline *BRCA1* pathogenic mutation; however, no associations could be found for *BRCA2* mutation carriers [25]. Neoadjuvant chemotherapy can significantly increase the density of CD8^+^ TILs, and their density has been associated with both PFS and OS [24].

In contrast to CD4^+^ and CD8^+^ TILs, CD45^+^ tumor-infiltrating leukocytes in ovarian cancer are a somewhat less researched area, and in many cases, the naive (CD45RA) and the memory (CD45RO) T cell isoforms are specifically investigated. CD45RO^+^ tumor-infiltrating leukocytes have been positively associated with survival [44,45]. Those patients with CD45RO^+^ tumors were also found to have longer survival [45,46], and the combination of CD8^+^ and CD45RO^+^ positivity was suggested as a better indicator for prognosis than the two markers alone [46]. Early- and advanced-stage tumors were reported to have different CD45RA^+^ tumor-infiltrating leukocytes (50% vs. 10.5%, respectively) [45]. Patients from a mesenchymal subtype (derived from immune gene expression subtyping in ovarian cancer) had the lowest tumor purity, a high leukocyte fraction, and a stromal fraction with the highest TGF-β response [47]. Furthermore, in patients with brain metastasis, it has been found that both the primary and brain metastasis samples contained approximately the same amount of CD45RO^+^ and CD8^+^ tumor-infiltrating immune cells [48], similar to that of the study investigating the omental metastasis samples detailed above [27].

In the current RROS, we confirmed that most of the included ovarian cancer patients (94.7%) had IHC positivity for CD4^+^, CD8^+^, and/or CD45^+^ immune cells. We were able to justify that strong CD4^+^ and CD45^+^ staining was the most advantageous in terms of patient survival. When investigated in a multivariate setting, CD4^+^ was the only marker of the three that had a significant effect on OS. However, we could not confirm the significant effect of tumor-infiltrating CD8^+^ T cells on patient survival. Moreover, we also demonstrated that CD4^+^ TILs were more common in the younger patients, hemoglobin and hematocrit values were lower in those with strong CD8^+^ staining, and the length of hospital stay was shorter with higher CD45^+^ staining. Our results, both the survival data and the associations with the other clinical parameters, fit perfectly with what is known from the literature. It should be noted, however, that in majority of the literature data, the presence of CD8^+^ TILs are generally positively associated with survival and results similar to ours are seen in only a minority of the literature.

A further goal of the current study was to investigate the gene expression profiles of the patients. Previous studies in the literature used several different commercially available gene expression profiling kits. The comparison of our results with previous data was limited to studies using the PanCancer^®^ gene expression profiling kits manufactured by NanoString, supplemented by studies in which similar comparisons were performed. Of the studies using any of the NanoString nCounter^®^ product line-up [12,13,14,15,49,50,51,52,53,54,55,56], the same PanCancer^®^ IO360^TM^ system used in our study was investigated in three [12,15,51]. In the study of James et al. [15], immunologic changes resulting from neoadjuvant chemotherapy exposure were investigated. They found that 10–10 DEGs were up- and down-expressed, and a significant decrease in cell proliferation, DNA damage repair, and epigenetics annotations, and a significant increase in the cytotoxicity and interferon signaling functions, were found. Furthermore, significant changes in the ‘proliferation and stress response’, ‘pro-tumorigenic signaling pathways’, and ‘immunoregulatory pathways’ were identified, all of which suggest that neoadjuvant chemotherapy can induce changes in immunostimulation and immunosuppression [15]. By comparing the significant DEGs of James’ study [15] with those of the current work, 36 of the tested genes were up- or down-expressed in both. The majority of the overlapping DEGs belonged to the ‘myeloid cell activity’ (*n* = 17), ‘immune cell localization to tumors’ (*n* = 13), ‘common signaling pathways’ (*n* = 10), and the ‘myeloid compartment’ (*n* = 9) annotation categories. Similarly, Jordan et al. [12] also compared pre- and post-neoadjuvant chemotherapy samples. They identified 69 significant DEGs, of which 12 overlapped with our results. Three of the overlapping DEGs belonged to each of the ‘antigen presentation’ (*MRC1*, *SOCS1*, *UBE2C*), ‘cell proliferation’ (*CCND2*, *CENPF*, *UBE2C*), ‘common signaling pathways’ (*DUSP1*, *DUSP5*, *SOCS1*), and ‘immune cell localization to tumors’ (*CDH5*, *CPA3*, *IFITM2*) annotations. Finally, the study of Rocconi et al. [51] investigated the relationship between the gene expression profile and Gemogenovatucel-T (an autologous tumor cell vaccine) treatment. In the current study, we compared patients with improved and poor survival (<12 months and >12 months, respectively), different ASA performance scores, different CD4^+^, CD8^+^, and CD45^+^ staining categories, and different T statuses. A total of 148 significant DEGs were found, of which the ‘myeloid cell activity’ (*n* = 61), ‘immune cell localization to tumors’ (*n* = 59), ‘killing of cancer cells’ (*n* = 37), ‘common signaling pathways’ (*n* = 35), ‘stromal factors’ (*n* = 28), ‘T cell priming and activation’ (*n* = 28), and ‘myeloid compartment’ (*n* = 27) annotations were the most common. Although not using NanoString, the study of Barnett et al. [26] compared tumors with high and low T_reg_ infiltrations. Using 200 DEGs, the authors predicted high vs. low T_reg_ infiltration tumors with 77% overall accuracy, and the antigen presentation pathway was the most affected of all investigated functional annotations [26]. Validation of the results of our gene expression profiling data was also performed using the KMplotter web application [19], and the same tendencies were justified, which further strengthened our results.

### Limitations

The current study has limitations. The sample size of the study was low, and gene expression analysis via NanoString could be performed only for an even lower, select number of patients. The selection of ovarian cancer cases for the NanoString analysis based on the prognosis might be biased by possible associations with other pathological and/or oncological parameters, such as by the correlation with tumor-infiltrating immune cells, subtypes, or whether the treatments of patients with different survival lengths were different. A further limitation was that the study population was not limited to a single ovarian cancer subtype, which increased the heterogeneity of the study. Moreover, the oncological treatment data of patients could not be obtained for every study participant, which might have introduced some bias in the survival analyses.

## 4. Materials and Methods

### 4.1. Patients and Study Design

A RROS was conducted with the inclusion of 57 histologically confirmed ovarian cancer patients, who were diagnosed between 2003 and 2017. Patients were operated on at the Department of Obstetrics and Gynecology, Saint Pantaleon Hospital, Dunaujvaros, Hungary. All patients signed a general informed consent form at the time of their operation, in which they all agreed to the anonymous data collection for scientific purposes. The study was approved by the Regional and Institutional Committee of Science and Research Ethics, Semmelweis University (SE TUKEB 133/2015), and the Medical Research Council (ETT-TUKEB 14383/2017). Routine pathological analysis of surgical specimens was performed at the Department of Pathology, Saint Pantaleon Hospital, Dunaujvaros, Hungary, while further analyses, including immunohistochemical and genetic testing, were carried out at the Department of Pathology, Forensic and Insurance Medicine, Semmelweis University, Budapest, Hungary.

### 4.2. Immunohistochemical Staining of OC Samples

Tissue microarrays (TMA) were composed from the FFPE samples with a systematic core punching algorithm using the Tissue Microarray Builder instrument (Histopathology Ltd., Pécs, Hungary). A total of 57 cores of 2 mm in diameter were taken from the main tumor mass. Immunohistochemical reactions were performed on 4 µm-thick sections cut from TMA blocks mounted on adhesive glass slides (SuperFrost UltraPlus from Gerhard Menzel Ltd., Braunschweig, Germany). We analyzed the following biomarkers in each section: CD4 (helper T cell marker), CD8 (cytotoxic T cell), and CD45 (leukocytes, LCA), in an automated immunostainer (Ventana Benchmark XT, Roche, Tucson, AZ, USA) using the solutions and settings as provided by the manufacturer. For CD45, both CD45RO and CD45RA were detected (Figure 6). The slides were digitalized with a Pannoramic P250beta slide scanner (3DHistech Ltd., Budapest, Hungary), and for the evaluation we used the Pannoramic Viewer with the support of the TMA and the Histoquant modules (3Dhistech Ltd., Budapest, Hungary). Immunoreactions were measured in the lymphocytes/leukocytes regarding all reactions and categorized as follows: negative, weak, moderate, and strong immunostaining were found if no (0%), 1–5%, 6–15%, or >15% of immune cells were stained, respectively. The percentage was determined as the density of infiltration according to the Salgado criteria, in line with the standardization and guidelines of TIL assessment, and adapted to IHC [57] (please see an extended and detailed description in the Appendix A). The reactions were quantified and calculated with computer-assisted image analysis by a histopathologist, using the QuPath (version 0.4.0) software environment, resulting in the number of positive cells per annotation, where the size of each annotation corresponded to the core cylinder’s surface of 3.14 mm^2^ [58].

### 4.3. NanoString nCounter PanCancer IO 360 Panel

The RNA of 22 tumors was obtained from FFPE tissue. Five 5-µm thick sections were cut from the FFPE blocks and, by following the manufacturer’s instructions, total RNA was obtained with the High Pure FFPE RNA Isolation Kit (Roche, Basel, Switzerland). RNA concentrations were measured using the Qubit 4 Fluorometer (Thermo Fisher Scientific, Waltham, MA, USA). The RNA samples with adequate concentrations were hybridized to the nCounter^®^ PanCancer IO 360^TM^ Gene Expression Panel (NanoString, Seattle, WA, USA), containing 770 genes, for 16 h using a thermocycler. The samples were transferred to the nCounter Prep Station (NanoString, Seattle, WA, USA) for further processing. The gene expression profiles of the samples were digitalized with the nCounter Digital Analyzer. The results were quantified using nSolver 4.0 Analysis Software (NanoString, Seattle, WA, USA).

### 4.4. Description of Clinicopathological Data

Pre-, peri-, and post-operative data were collected. Complete blood count and tumor marker measurements were completed at the Central Laboratory of Saint Pantaleon Hospital, Dunaujvaros, Hungary. Age, weight, medical history data, the ASA performance score [18], the type of tumor removal surgery, and results of routine pathological analysis were retrospectively collected from the hospital information system of Saint Pantaleon Hospital, Dunaujvaros, Hungary. Oncological treatment of patients occurred either at the Division of Oncology, Department of Internal Medicine and Oncology, Semmelweis University, Budapest, Hungary, or at another institution not participating in this study, according to the patients’ preferences. Where possible, all details of chemotherapy (medication, number of cycles, etc.) were collected. OS was calculated from the date of diagnosis to the date of death from any cause or until the last follow-up date.

### 4.5. Data Collection from the Kaplan–Meier Plotter Web Application

The Kaplan–Meier Plotter web application [19] (developed by Győrffy et al. and available online at https://kmplot.com/analysis/index.php?p=service&cancer=ovar; access date: 20 July 2023) is a web application used to discover and validate survival biomarkers in various tumorous diseases, including ovarian cancer. The tool provides a wide variety of gene/mRNA count data, with corresponding survival times and censoring data of the same patients. The OS data were obtained by manually selecting the genes of interest and splitting the measurement data by the median without any further changes to the default settings. From the results presented by the application, the hazard ratio (HR) and its 95% confidence interval (CI), the *p*-value from the log rank test, and the median OS of the low- and high-expression groups were collected.

### 4.6. Statistical Analysis

Data were analyzed within the R for Windows environment (version 4.3.1, R Foundation for Statistical Computing, Vienna, Austria). Continuous and count data comparisons were performed using the Kruskal–Wallis ANOVA with *p*-value-adjusted pairwise comparisons using the Wilcoxon rank sum exact test and Fisher exact tests, respectively. Correlation analysis was performed using the Spearman correlation. Cox regression and the log rank test were used to assess survival data (R package survival, version 3.5–5) and plotted using the survminer R package (version 0.4.9). A *p*-value < 0.05 was considered statistically significant. Continuous, survival, and count data were expressed as the mean ± standard deviation, the HR with a 95% CI, and the number of observations (percentage), respectively.

NanoString data were analyzed using the RUVSeq method [59] (R package RUVSeq, version 1.34.0 [60]). In short, after the normalization of the count data with RUVSeq, differential expression analysis (R package DESeq2, version 1.40.2 [61]) and gene set enrichment analysis (GSEA) were performed. The results were drawn using volcano plots and heatmaps with the ggplot2 (version 3.4.2 [62]) and ComplexHeatmap (version 2.16.0 [63]) R packages, respectively.

## 5. Conclusions

In our RROS, we found that tumor-infiltrating lymphocyte counts are associated with peculiar gene expression patterns and bear prognostic information in ovarian cancer: CD4^+^ and CD45^+^ immune cell infiltration showed significant predictive power for overall survival. The gene *SOX11* was identified and validated in an independent dataset as a prognostic marker in ovarian cancer.

## Figures and Tables

**Figure 1 ijms-24-13684-f001:**
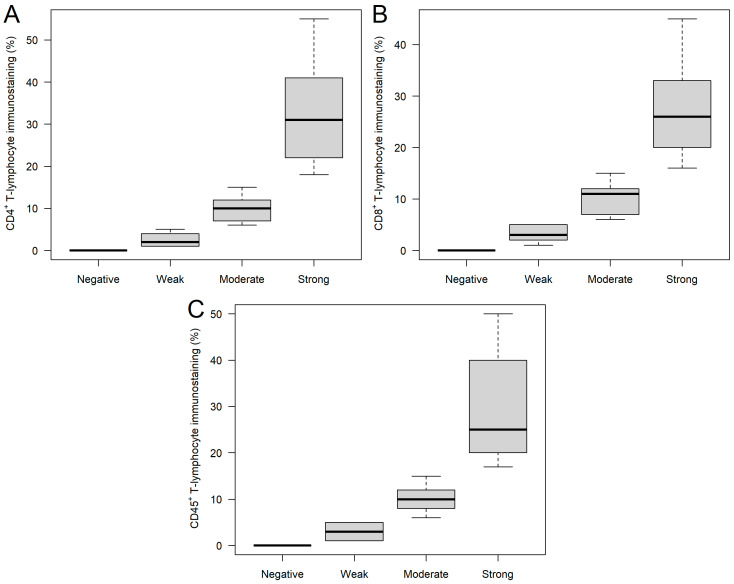
CD4^+^ (**A**), CD8^+^ (**B**), and CD45^+^ (**C**) immunostaining of ovarian tumor samples. Based on the percentage of stained tumor-infiltrating immune cells, the following categories were created: negative, weak, moderate, and strong staining were found if no (0%), 1–5%, 6–15%, or >15% of the T cells were stained.

**Figure 2 ijms-24-13684-f002:**
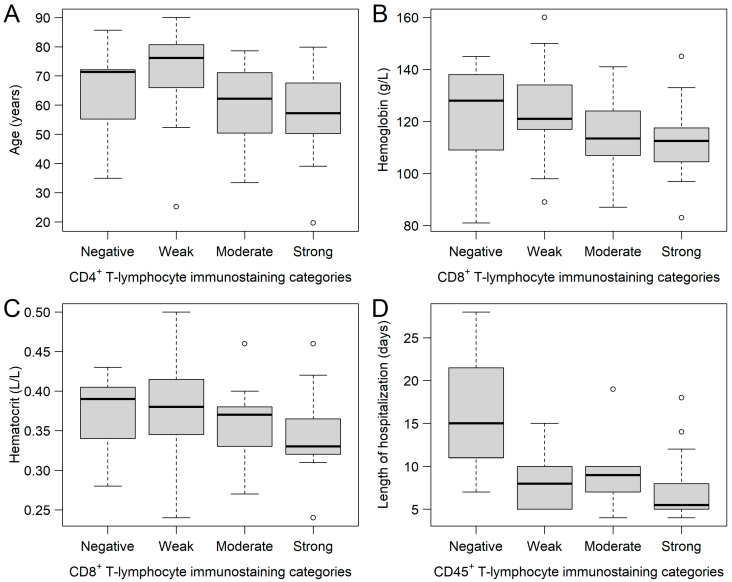
Relationships between the CD4^+^, CD8^+^, and CD45^+^ immunostaining sub-cohorts and various clinical parameters. (**A**) Patients with moderate and strong CD4^+^ T-lymphocyte immunostaining were significantly younger, compared to those with weak staining results. Hemoglobin (**B**) and hematocrit (**C**) values of patients were the lowest in those with strong CD8^+^ staining. Length of hospitalization (**D**) was the shortest in those patients with strong CD45^+^ staining. Hollow circles represent outliers (greater/lower 1.5 times the interquartile range above/below the upper/lower quartile).

**Figure 3 ijms-24-13684-f003:**
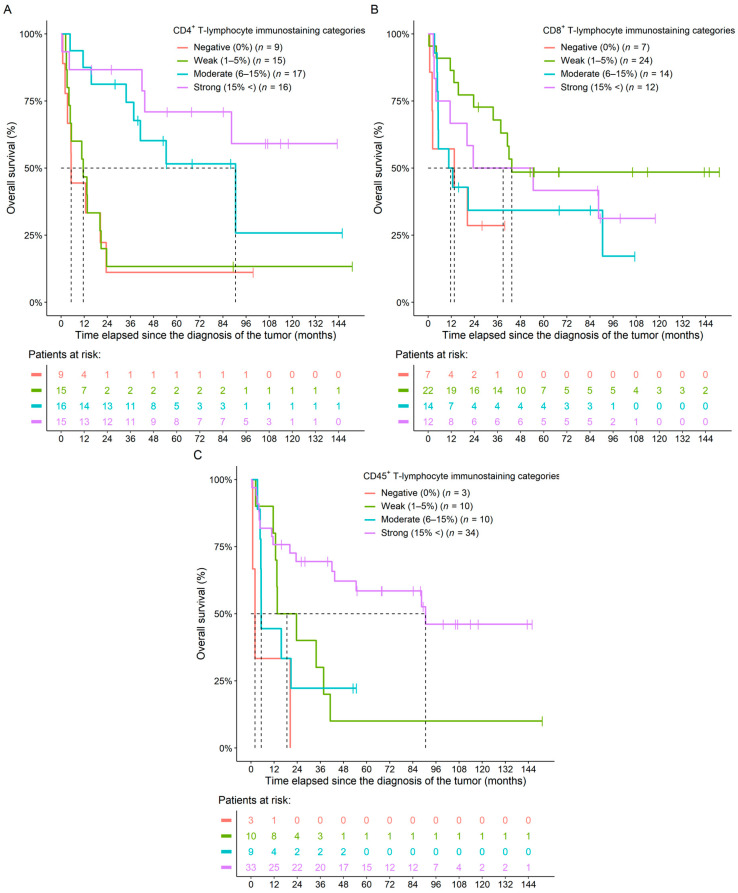
Naive Kaplan–Meier curves of the CD4^+^ (**A**), CD8^+^ (**B**), and CD45^+^ (**C**) immunostaining categories of ovarian cancer patients. Colors within the bottom tables represent the same cohorts as detailed in the figure legends.

**Figure 4 ijms-24-13684-f004:**
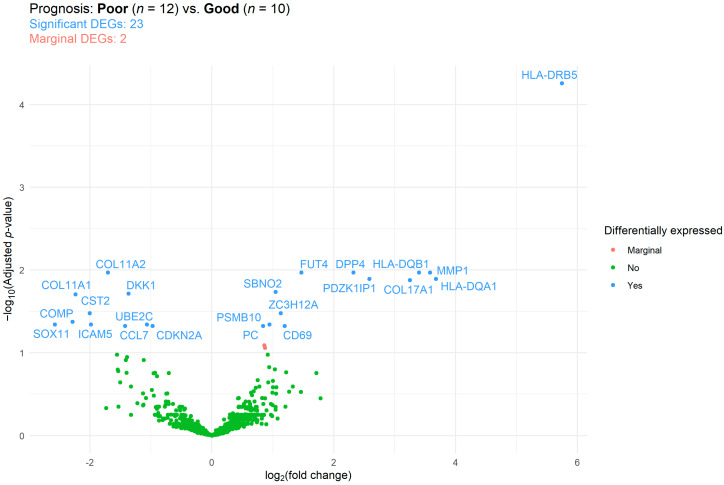
Differentially expressed genes (DEGs) between those ovarian cancer patients with good and poor prognosis (survival time < 1 and >1 years, respectively). The false discovery rate method was used for *p*-value adjustment. Reference category: patients with good prognosis.

**Figure 5 ijms-24-13684-f005:**
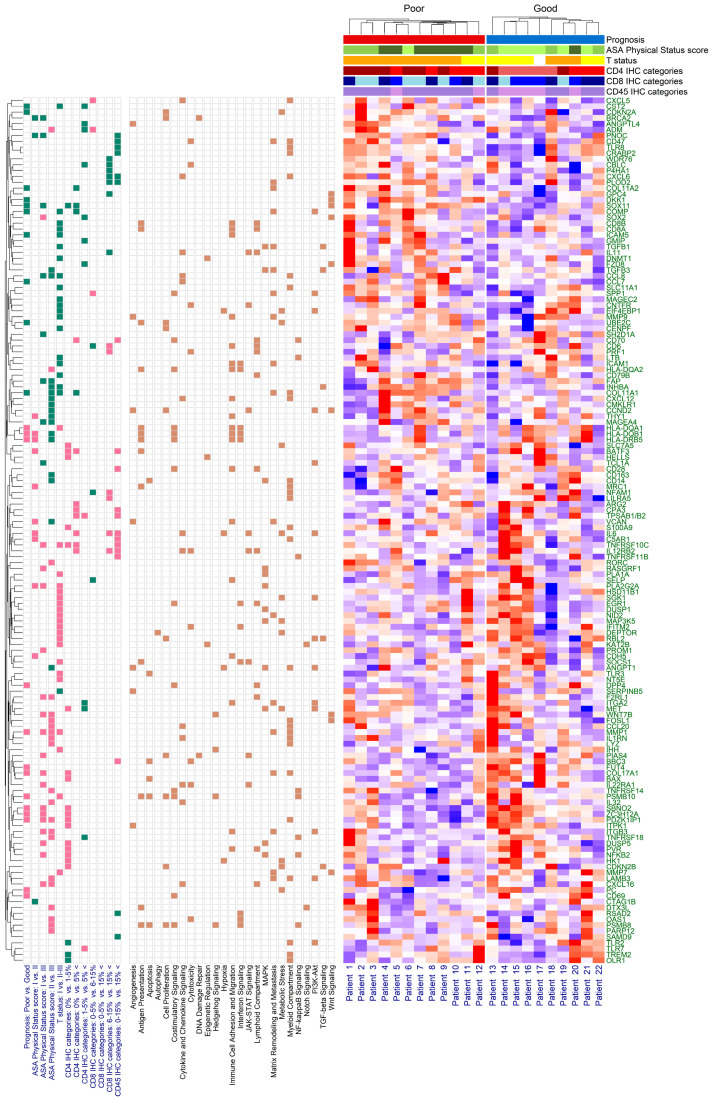
Heatmap of significantly different gene expressions. The green and pink boxes represent the down-expression and the up-expression of genes, respectively. The brown box shows cancer-immunity cycle enrichment annotation information of the differentially expressed genes. The American Society of Anesthesiologists (ASA) performance scores are represented as: I (light green), II (green), and III (dark green); T status as: I (yellow) and II–III (orange); CD4 IHC categories as: 0% (light red), 1–5% (red), and 5%< (dark red); CD8 IHC categories as: 0–5% (light blue), 6–15% (blue), and 15%< (dark blue); CD45 IHC categories as: 0–15% (light purple) and 15%< (purple).

**Figure 6 ijms-24-13684-f006:**
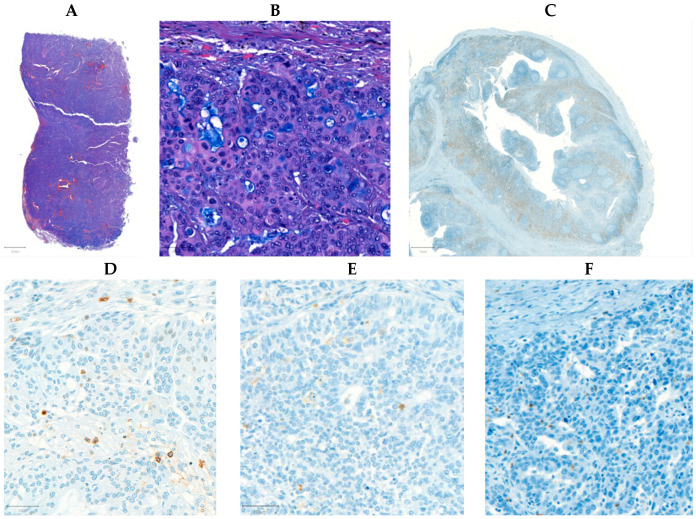
Microscopic images depicting a high-grade serous ovarian carcinoma. The hematoxylin and eosin-stained tissue is displayed at 0.6× (**A**) and 10× (**B**) magnification. A positive control for CD4 taken from the appendix of one of the surgically resected cases at 2× magnification (**C**). Stains for CD45 (**D**), CD8 (**E**), and CD4 (**F**) at 10× magnification.

**Table 1 ijms-24-13684-t001:** Anamnestic and clinicopathological data of ovarian cancer patients. Continuous and count data are reported as mean ± standard deviation and the number of observations (percentage), respectively.

Parameter	All Study Participants (*n* = 57)
Age (year)	63.02 ± 15.67
Weight (kg)	73.14 ± 13.92
No. of births	1.86 ± 0.93
No. of abortions	0.44 ± 0.87
Preoperative laboratory results	
- Hemoglobin (g/L)	118.98 ± 17.51
- Hematocrit (L/L)	0.37 ± 0.06
- Platelet count (10^9^/L)	334.95 ± 125.73
- Carcinoembryonic antigen (ng/mL)	103.47 ± 441.21
- Carbohydrate antigen 125 (U/mL)	1118.26 ± 1567.83
Symptoms	
- Duration (months)	4.39 ± 7.21
- Abdominal pain	31 (54.39%)
- Increased abdominal circumference	14 (24.56%)
- Bloating	6 (10.53%)
- Weight loss	15 (26.32%)
- Shortness of breath	6 (10.53%)
- Bowel changes	7 (12.28%)
Surgery details	
- Total abdominal hysterectomy	41 (71.93%)
- Bilateral salpingo-oophorectomy	46 (80.70%)
- Unilateral salpingo-oophorectomy	7 (12.28%)
- Tumor reduction surgery only	4 (7.02%)
- Omentectomy	46 (80.70%)
- Length of hospitalization (days)	8.07 ± 4.50
- Transfusion (unit)	0.68 ± 1.68
ASA performance score (I:II:III:IV) [18]	14:20:20:3 (24.56%:35.09%:35.09%:5.26%)
Histology	
- Endometrioid carcinoma	6 (10.53%)
- Serous cystadenocarcinoma	29 (50.88%)
- Mucinous cystadenocarcinoma	10 (17.54%)
- Clear cell carcinoma	4 (7.02%)
- Other types (including mixed types)	8 (14.04%)
Medical history	
- Hypertension	28 (49.12%)
- Type 2 diabetes mellitus	15 (26.32%)
- Major cardiovascular events	7 (12.28%)
- Appendectomy	4 (7.02%)
- Cholecystectomy	7 (12.28%)

ASA: American Society of Anesthesiologists.

**Table 2 ijms-24-13684-t002:** Cox regression model results investigating the univariate and multivariate effects of raw CD4^+^, CD8^+^, and CD45^+^ tumor-infiltrating immune cell percentages over the overall survival of ovarian cancer patients. Interpretation of these data: for every 1% increase, the risk for shorter survival increases/decreases with the HR detailed in the table, e.g., a 0%, 20%, and 80% CD4^+^ infiltration is associated with a 0.9459^0^ (HR = 1), 0.9459^20^ (HR = 0.3288), and 0.9459^80^ (HR = 0.0117) risk for shorter survival.

Parameter	Univariate HR (95% CI)	Univariate *p*-Value	Multivariate HR (95% CI)	Multivariate *p*-Value
CD4^+^ T-lymphocytes (%)	0.9459 (0.9121–0.9810)	0.0028	0.9403 (0.8869–0.9970)	0.0392
CD8^+^ T-lymphocytes (%)	1.0050 (0.9754–1.0350)	0.7480	0.9967 (0.9548–1.0400)	0.8801
CD45^+^ leukocytes (%)	0.9685 (0.9422–0.9954)	0.0221	1.0073 (0.9544–1.0630)	0.7913

CI: confidence interval; HR: hazard rate. Note: The multivariate model included only the CD4^+^, CD8^+^, and CD45^+^ percentage data, and no further clinical parameter was included.

**Table 3 ijms-24-13684-t003:** Hazard rates and *p*-values of univariate survival models investigating the effect of CD4^+^, CD8^+^, and CD45^+^ immunostaining categories over the overall survival of ovarian cancer patients. The corresponding naive Kaplan–Meier curves can be seen in Figure 3.

Immunostaining Sub-Cohorts	CD4^+^	CD8^+^	CD45^+^
Negative (ref.) vs. weak	0.8689 *p* = 0.7550	0.3206 *p* = 0.0389	0.3105 *p* = 0.0852
Negative (ref.) vs. moderate	0.2541 *p* = 0.0072	0.6884 *p* = 0.5021	0.3474 *p* = 0.1317
Negative (ref.) vs. strong	0.1543 *p* = 0.0013	0.4932 *p* = 0.2253	0.1132 *p* = 0.0010
Weak (ref.) vs. moderate	0.2924 *p* = 0.0074	2.1473 *p* = 0.0830	1.1188 *p* = 0.8254
Weak (ref.) vs. strong	0.1776 *p* = 0.0013	1.5385 *p* = 0.3550	0.3647 *p* = 0.0194
Moderate (ref.) vs. strong	0.6072 *p* = 0.3833	0.7165 *p* = 0.4850	0.3260 *p* = 0.0174

Ref: reference category.

**Table 4 ijms-24-13684-t004:** Differentially expressed genes that were significantly up- (+) or down-expressed (−) in at least 4 of the comparisons.

Comparison	*BATF3*	*COL11A1*	*IL6*	*MMP1*	*PDZK1IP1*	*SOX11*	*TNFRSF10C*
Prognosis: Good (ref.) vs. Poor		–		+	+	–	
ASA score: I (ref.) vs. II	+		+				
ASA score: I (ref.) vs. III				+	+	+	+
ASA score: II (ref.) vs. III		–		+	+		
T status: I (ref.) vs. II-III		–		+			+
CD4^+^: 0% (ref.) vs. 1–5%	+				+	–	+
CD4^+^: 0% (ref.) vs. 5%<	+	–	+			–	+
CD8^+^: 6–15% (ref.) vs. 15%<			+				
CD45^+^: 0–15% (ref.) vs. 15%<	+		+				+

*BATF3*: basic leucine zipper ATF-like transcription factor 3; *COL11A1*: collagen type XI alpha 1 chain; *IL6*: interleukin 6; *MMP1*: matrix metallopeptidase 1; *PDZK1IP1*: PDZK1 interacting protein 1; *SOX11*: sex-determining region Y-box transcription factor 11; *TNFRSF10C*: tumor necrosis factor receptor superfamily member 10c; ref: reference category.

**Table 5 ijms-24-13684-t005:** Validation comparisons of the results of the current study, compared to the Kaplan–Meier Plotter web application [19]. *p*-Values were obtained from log rank tests. It is notable that the Kaplan–Meier Plotter web application contains data on ovarian cancer patients with serous and endometrioid types only. Therefore, from our database, only those patients with similar histology were included.

Gene	Hazard Rate	95% Confidence Interval	*p*-Value	Median Survival Time (Months)	
				Lower Expression Group	Higher Expression Group
A: Data obtained from the measurements of the current study					
*COL11A1*	1.1588	0.3092–4.3430	0.8268	8.26	4.01
*COL17A1*	0.4503	0.1111–1.8257	0.2522	4.37	11.40
*COMP*	2.4723	0.6520–9.3754	0.1696	75.52	4.73
*CTAG1B*	0.7193	0.1923–2.6908	0.6229	4.99	5.13
*HLA-DQA1*	0.5508	0.1365–2.2225	0.3954	4.93	11.40
*HLA-DQB1*	0.5508	0.1365–2.2225	0.3954	4.93	5.26
*HLA-DRB5*	0.3528	0.0873–1.4263	0.1272	4.22	5.26
*IL6*	0.9211	0.2449–3.4637	0.9031	4.99	5.13
*ITGB3*	2.0631	0.5501–7.7382	0.2728	75.58	3.32
*LYZ*	0.4036	0.1003–1.6245	0.1869	3.53	5.26
*MAGEC2*	0.7766	0.2068–2.9159	0.7072	4.99	5.13
*MMP1*	0.9489	0.2519–3.5747	0.9382	4.63	5.13
*PDZK1IP1*	0.3304	0.0792–1.3782	0.1129	4.37	11.40
*PLA2G2A*	0.5690	0.1510–2.1444	0.3990	4.22	5.26
*SOX2*	1.5048	0.4025–5.6254	0.5408	59.53	4.73
*SOX11*	8.4100	1.6237–43.5603	0.0032	78.65	3.32
*TNFRSF18*	0.2826	0.0665–1.2010	0.0716	3.66	11.40
B: Data obtained from the Kaplan–Meier Plotter web application [19]					
*COL11A1*	1.30	1.14–1.48	<0.0001	49.47	40.54
*COL17A1*	0.91	0.80–1.03	0.1400	44.80	45.63
*COMP*	1.18	1.04–1.34	0.0110	48.00	41.00
*CTAG1B*	1.02	0.90–1.16	0.7200	45.63	44.77
*HLA-DQA1*	0.96	0.84–1.09	0.4900	45.13	45.53
*HLA-DQB1*	0.95	0.84–1.08	0.4500	45.23	45.17
*HLA-DRB5*	0.89	0.78–1.01	0.0820	43.00	48.00
*IL6*	1.07	0.94–1.22	0.3100	45.63	44.80
*ITGB3*	0.93	0.82–1.06	0.2600	44.70	45.77
*LYZ*	0.91	0.80–1.04	0.1500	43.00	46.52
*MAGEC2*	0.96	0.84–1.09	0.5000	44.53	45.77
*MMP1*	1.08	0.95–1.23	0.2500	45.53	45.00
*PDZK1IP1*	0.86	0.75–0.97	0.0180	41.89	48.00
*PLA2G2A*	1.04	0.91–1.18	0.5400	45.40	45.00
*SOX2*	1.01	0.82–1.24	0.9300	45.00	41.89
*SOX11*	1.25	1.10–1.42	0.0008	48.27	41.00
*TNFRSF18*	0.93	0.76–1.14	0.4800	43.00	45.00

*COL11A1*: collagen type XI alpha 1 chain; *COL17A1*: collagen type XVII alpha 1 chain; *COMP*: cartilage oligomeric matrix protein; *CTAG1B*: cancer/testis antigen 1B; *HLA-DQA1*: major histocompatibility complex, class II, DQ alpha 1; *HLA-DQB1*: major histocompatibility complex, class II, DQ beta 1; *HLA-DRB5*: major histocompatibility complex, class II, DR beta 5; *IL6*: interleukin 6; *ITGB3*: integrin subunit beta 3; *LYZ*: lysozyme; *MAGEC2*: MAGE family member C2; *MMP1*: matrix metallopeptidase 1; *PDZK1IP1*: PDZK1 interacting protein 1; *PLA2G2A*: phospholipase A2 group IIA; *SOX2*: sex-determining region Y-box transcription factor 2; *SOX11*: sex-determining region Y-box transcription factor 2; *TNFRSF18*: tumor necrosis factor receptor superfamily member 18.

## Data Availability

The datasets used and/or analyzed during the current study are available from the corresponding author upon reasonable request.

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
