# Peer review of "High Tumor-Infiltrating Lymphocyte Count Is Associated with Distinct Gene Expression Profile and Longer Patient Survival in Advanced Ovarian Cancer"

_ijms, 2023, doi:10.3390/ijms241813684_

Round 1

Reviewer 1 Report

The manuscript “High tumor infiltrating lymphocyte count is associated with distinct gene expression profile and longer patient survival in advanced ovarian cancerby Andras Jozsef Barna and co-authors to conduct a retrospective real-life observational study including 57 advanced ovarian cancer patients. Immunohistochemistry for CD4+, CD8+ and CD45+ was used to assess tumor-infiltrating leukocytes (TIL). A higher number of CD4+ (p = 0.0028) and CD45+ (p = 0.0221) immune cells within the tumor microenvironment were associated with longer overall survival of patients. In a multivariate setting, CD4+ T cell infiltration predicted overall survival (p = 0.0392). Twenty-three differentially expressed genes - involved in antigen presentation, costimulatory signaling, matrix remodeling, metastasis formation, and in myeloid cell activity - were found when comparing the prognostic groups. It was found that TIL counts are associated with peculiar gene expression patterns and bear prognostic information in ovarian cancer. PDZK1IP1 expression emerged and was validated as a predictive marker for OS. However, there are concerns that must be taken into account before the work can be reconsidered for publication.

1.      The IHC photos of CD4+, CD8+ and CD45+ should be provided. How to calculate the cut-off point about CD4+, CD8+ and CD45+? The detailed describe should be added.

2. The correlation between CD4+, CD8+ and CD45+ and clinical parameters should be analyzed. If these patients received chemotherapy?

 Extensive editing of English language required

Reviewer 2 Report

The authors tackle a highly relevant aspect of gynecological oncology, analyze the impact of tumor-infiltrating lymphocytes in ovarian cancer, and highlight certain impacts on the overall gene expression in the particular tumor tissue. The manuscript is well written, and the scientific question is addressed properly. However, there are some aspects that should be clarified before being suitable for publication.

1. The overall status of chemotherapy treatment is not mentioned. The impact of chemotherapy on the analysis, either in terms of quality or time of the treatment, might have a huge impact on the data, especially in the context of a smaller cohort. Since the authors also refer to a lot of literature concerning chemotherapy in their discussion, it is necessary to provide this clinical data and, if applicable, include it in the analysis.

2. Figure 1: The authors show a broad variety of TIL concentrations in their cohort. However, do the different immune-cell populations correlate within the samples? At least CD45 and CD4/CD8 should correlate well and would be an additive internal control for the performed staining. Additionally, a determination of the total TIL amount by HE staining could be a good addendum to the analysis since it gives a good overview of the immune status of the particular patient. Moreover, it is not clear which compartment concerning immune cell localization was analyzed. The authors should clarify whether the TILs were quantified in stroma, tumor, edges, or total. Furthermore, is there any correlation between TILs and the different histological subtypes ?

3. Table 2: How was the multivariable survival model constructed, and which parameters have been included in the analysis? Concerning the small sample size, this should be stated. Moreover, if applicable, which cut-offs have been used for the individual immune cell staining? Additionally, an HR of 0.95-0.98 might be a rather weak effect if the base model refers to an HR of 1. So the results should be interpreted with caution.

4. The selection of ovarian cancer cases based on the prognosis reflects a clinically relevant parameter, and is certainly of interest. However, it remains unclear if the cases with good prognosis correlate with the determined TILs, which subtype was selected and if the treatment was similar. It would be good to provide this information.

5. To my understanding, the authors have performed an initial nanostring analysis based on good vs. bad survival. Afterward, the same dataset was stratified according to certain immune cell contents. It remains rather counterintuitive why the impacts on expression in Table 4 are contradictory for many of the identified genes (CLO11A1, MMP1, PDZK1IP1). The authors should address this in the discussion. If the increased accumulation of immune cells fosters a better prognosis, why are the favorable genetic factors inversely regulated?

6. Table 5. Using in silico data for validation is a great tool and a good addition to the study. However, the KM Plotter data consist mainly of serous ovarian cancers, and the cancers in this study are of mixed subtypes. A direct comparison might be appropriate in terms of this complex topic. However, it should be stated.

7. A conclusive chapter and interpretation of the study data could be of great interest. How do the authors interpret their data in the context of PARP inhibition and ICB, for example? Do they see potential for therapy decision-making and prediction?

8. The discussion summarizes a lot of literature concerning the prognostic value of TILS in ovarian cancer and strengthens the results of this study. However, the cited references tackle a lot of topics that have not been addressed in the study and could be removed or interpreted better. e.g., the expression in metastatic tissue, the roles of regulatory T-cells, germ line mutations.  Additionally, since the authors state a differential effect between CD45RO and CD45RA, which population did they detect with their staining ? If both were detected, this could be stated in the MM section. The chapter about the effects of adjuvant and/or neoadjuvant chemotherapy could be better related to the study by once again stating the therapy status in this project.

Round 2

Reviewer 1 Report

The revised manuscript “High tumor infiltrating lymphocyte count is associated with distinct gene expression profile and longer patient survival in advanced ovarian cancerhave adequately addressed my previous concerns and the paper is now acceptable for publication.

Reviewer 2 Report

The authors have adressed my questions and provided good solutions for my concerns.

Thank you for the revised manuscript.